# *In Vitro* and *In Vivo* Assessment of Dietary Supplementation of Both Natural or Nano-Zeolite in Goat Diets: Effects on Ruminal Fermentation and Nutrients Digestibility

**DOI:** 10.3390/ani11082215

**Published:** 2021-07-27

**Authors:** Amr El-Nile, Mahmoud Elazab, Hani El-Zaiat, Kheir El-Din El-Azrak, Alaa Elkomy, Sobhy Sallam, Yosra Soltan

**Affiliations:** 1Department of Livestock Research, Arid Land Cultivation Research Institute, City of Scientific Research and Technological Applications, New Borg El-Arab, Alexandria 21934, Egypt; amr_elneel88@yahoo.com (A.E.-N.); melazab@srtacity.sci.eg (M.E.); alaa_elkomy@yahoo.com (A.E.); 2Department of Animal and Fish Production, Faculty of Agriculture, Alexandria University, Alexandria 21545, Egypt; hm_elzaiat@yahoo.com (H.E.-Z.); kheir_elazrak@yahoo.com (K.E.-D.E.-A.); uosra_eng@yahoo.com (Y.S.); 3Department of Animal and Veterinary Sciences, College of Agricultural and Marine Sciences, Sultan Qaboos University, P.O. Box 34, Al-Khod 123, Oman; 4Faculty of Desert and Environmental Agriculture, Matrouh University, Matrouh 51512, Egypt

**Keywords:** zeolite, nano-zeolite, *in vitro* gas production, digestibility, goat, methane emission, clay minerals

## Abstract

**Simple Summary:**

Increasing fibrous feed digestibility while reducing methane (CH_4_) emission through manipulating rumen fermentation patterns to improve animal performance is the most critical challenge in the animal nutrition field. Nanotechnology has revolutionized the commercial application of nano-sized minerals in medicine, engineering, information, environmental technology, pigments, food, electronics appliances, biological and pharmaceutical applications, and many more. Therefore, animal nutrition scientists also resorted to using minerals and clays such as zeolite with different forms in feeding animals and evaluate this additive in animal performance. The natural zeolite clay is known for its high cation exchange capacity and adsorption characteristics that can modify ruminal fluid viscosity and binding capacity with ammonia (NH_3_-N). After evaluating the addition of zeolite *in vivo* and *in vitro*, results indicated that zeolite (natural and nano forms) maintained rumen pH, increased protozoa numbers, and improved propionate production. Medium supplementation level of the natural form of zeolite at 20 g/kg dry matter (DM) was the most efficient dose in reducing CH_4_ production, while the zeolite nano-form supplemented at 0.4 g/kg DM was the most effective dose in improving the organic matter (OM) degradability and reducing the NH_3_-N concentration compared to the control.

**Abstract:**

This study aimed to evaluate *in vitro* and *in vivo* dietary supplementation with different levels of natural or nano-zeolite forms on rumen fermentation patterns and nutrient digestibility. In the *in vitro* experiment, a basal diet (50% concentrate: 50% forage) was incubated without additives (control) and with natural zeolite (10, 20, 30 g/kg DM) or nano-zeolite (0.2, 0.3, 0.4, 0.5, 1.0 g/kg DM) for 24 h to assess their effect on ruminal fermentation, feed degradability, and gas and methane production using a semi-automatic system of *in vitro* gas production (GP). The most effective doses obtained from the *in vitro* experiment were evaluated *in vivo* using 30 Barki goats (26 ± 0.9 SE kg body weight). Goats were allocated into three dietary treatments (*n* = 10/treatment) as follows: control (basal diet without any supplementations), natural zeolite (20 g/kg DM diet), and nano-zeolite (0.40 g/kg DM diet). The *in vitro* results revealed that only the nano-zeolite supplementation form quadratically (*p*
*=* 0.004) increased GP, and the level of 0.5 g/kg DM had the highest GP value compared to the control. Both zeolite forms affected the CH_4_ production, linear, and quadratic reductions (*p* < 0.05) in CH_4_ (mL/g DM), consistent with linear increases in truly degraded organic matter (TDOM) (*p* = 0.09), and propionate molar proportions (*p* = 0.007) were observed by nano zeolite treatment, while the natural form of zeolite resulted in a linear CH_4_ reduction consistent with a linear decrease (*p* = 0.004) in NH_3_-N, linear increases in TDOM (*p* = 0.09), and propionate molar proportions (*p* = 0.004). Results of the *in vivo* experiment demonstrated that the nutrient digestibility was similar among all treatments. Nano zeolite enhanced (*p* < 0.05) the total short-chain fatty acids and butyrate concentrations, while both zeolite forms decreased (*p* < 0.001) NH_3_-N compared to the control. These results suggested that both zeolite supplementation forms favorably modified the rumen fermentation in different patterns.

## 1. Introduction

The application of feed additives in ruminant rations is one solution to improve the animal’s performance via manipulating ruminal fermentation patterns and improving nutrients utilization. Microbial fermentation of the dietary organic matter results in loss of gross energy and nitrogen. Enteric CH_4_ emission in ruminants represents a loss of up to 15% of gross energy of feeds; also, 75–85% of the nitrogen consumed by ruminants is excreted in the feces and urine [1]. Therefore, enhancing fibrous feed digestibility, reducing CH_4_ emission, and nitrogen excretion by ruminants have to improve their performance [2].

Natural zeolite clay is composited of crystalline aluminosilicates and characterized by a high cation exchange capacity, high sorbent property that can modify ruminal fluid viscosity and binding capacity with NH_3_-N; therefore, it has been extensively used as a potential feed additive [3]. It also can capture ammonium ions, reducing the rate of their release and absorption from the rumen wall, and act as adsorbents for mycotoxins [4]. Besides, clinoptilolite of zeolite can enhance microbial ruminal fermentation by regulating ruminal pH to act as a pH-buffering agent [5].

The literature reported that zeolite supplementation levels had been examined ranging from 1% to 9% of DM of ruminant diets [6,7,8]. Dietary supplementation with zeolite clay exhibited positive effects on nutrients digestion and growth performance of sheep [9]. Furthermore, zeolite positively affected animal health status and performance due to its characteristic sorbent properties that modify the ruminal environment [10,11]. Nanoclays and other nano-particles have been shown to specifically absorb mycotoxins through the gastrointestinal tract of ruminants [12]. Nanotechnology is one of the most promising applications of the twenty-first century. It can create new materials with unique properties, which change the physical and chemical characteristics of the molecules/element to have the potential to revolutionize agriculture sectors and has given birth to the new area of agro-nanotechnology, particularly in livestock production. Size reduction of materials to the nano range can increase their adsorption, absorption, and cation exchange capacity [13]. Comparative research studies of nano and natural zeolite supplementations on rumen fermentation patterns and nutrient digestibility are limited. Therefore, we hypothesized that the effects of nano zeolite on ruminal microbial activity might differ from its natural form. Therefore, the objective of this study was to investigate the *in vitro* dose–response effects of natural and nano-zeolite supplementations on ruminal antimethanogenic activity, fermentation end-products, and nutrient degradation. The most effective doses of both zeolite forms were evaluated *in vivo* for ruminal fermentation characteristics and nutrient digestibility.

## 2. Materials and Methods

The study was carried out at the Advanced Laboratory of Animal Nutrition and experimental farm Faculty of Agriculture, Alexandria University and Laboratory of Livestock Research Department of Arid Land Cultivation Research Institute, the City of Scientific Research and Technological Applications, Alexandria. All procedures following protocols were approved and authorized by the Institutional Animal Care and Use Committee of the Alexandria University (ALEXU-IACUC/08-19-05-14-2-22).

### 2.1. Experimental Feed Additives

Natural zeolite was commercially purchased from A & O trading company, Giza, Egypt. Zeolite is composed of a microporous arrangement of silica and alumina tetrahedra (Clinoptilolite) with general formula (Ca, K_2_, Na_2_, Mg)_4_ Al_8_ Si_40_ O_96_. 24H_2_O. The chemical composition and physical properties of zeolite in its natural form are according to the zeolite datasheet by the A & O trading company (Table 1).

The nano-zeolite powder was prepared mechanically by a high-energy planetary ball mill (Retsch PM, Germany) [14]. The mechanical route was performed in a period of 6 h with a reverse rotation speed of 300 rpm and vial rotation speed of 600 rpm with a ball to powder ratio of 9:1 mass/mass. The particle size of the obtained nano zeolite was measured by N**_5_** submicron particle size analyzer (BECKMAN COULTER, Brea, CA, USA), with a range of 3 nm–5 µm of particle size.

To detect the distribution size and shape of zeolite nano-particles, the scanning electron microscope (SEM; Jeol JSM-6360 LA, 3-1-2 Musashino, Akishima, Tokyo, Japan) and transmission electron microscope (TEM; JEOL JEM-2100, 3-1-2 Musashino, Akishima, Tokyo, Japan) were used to provide three-dimensional images, which are very useful for understanding the morphological characters of the tested nanoparticles [15]. The sample was coated with gold to improve the imaging of the sample. The SEM was operated at a vacuum of the order of 10, and the accelerating voltage of the microscope was kept in the range of 10–20 kV (Figure 1). The TEM nano-particles’ shape and size were prepared by dropping approximately 10–15 µL of a dilute sample of ZnO-NPs on the top of the carbon-coated copper grid and left in the hood to dry (Figure 2). The particle size mean was 60.2 nm of the nano zeolite.

To identify the functional groups of the prepared nano-zeolite form, the Fourier Transform Infra-Red Spectroscopy (FTIR) analysis was performed using an infrared spectrometer (Shimadzu FTIR-8400S, Nakagyo-ku, Kyoto, Japan) by employing the KBr pellet technique [16], as shown in Figure 3.

The surface charge of the nano-zeolite was measured by zeta potential analysis using a Malvern ZETASIZER Nano series (Malvern, Worcestershire, England, United Kingdom) [17], under the following circumstances: temperature (°C) 25.0, count Rate (kcps) 347.4, measurement position (mm) 2.00, and attenuator 7.00. The zeta potential of the prepared nano-zeolite was −5.85 (mv), zeta deviation and conductivity were 63.8 (mV) and 0.00165 (mS/cm), respectively, as presented in Figure 4.

### 2.2. Basal Diet

The experimental basal diet (used in the *in vitro* and *in vivo* experiments) consisted of (g/kg DM) 500 g concentrate and 500 g berseem hay (*Trifolium alexandrinum*) formulated as a total mixed ration (TMR) diet to meet the nutrient requirements of lactating goats [18]. The AOAC [19] analytical procedures were used for dry matter (DM), organic matter (OM), crude protein (CP as 6.25 × N; by Kjeldahl technique), and ether extract (EE). Cell wall ingredients (neutral detergent fiber (NDF), acid detergent fiber (ADF), and lignin contents (ADL)) were determined sequentially by an Ankom 200 fiber analyzer unit (ANKOM Technology Corporation, Macedon, NY, USA) and expressed exclusive of residual ash as described by Van Soest et al. [20]. Concentrations of hemicellulose were calculated as NDF—ADF, and cellulose as ADF—ADL.

The major ingredients and chemical composition of the experimental diet are presented in Table 2.

### 2.3. The In Vitro Experiment

The experimental treatments consisted of control (basal diet without supplementation), five supplemental doses of the nano-zeolite (0.2, 0.3, 0.4, 0.5, and 1 g/kg DM basal diet), and three doses of the natural zeolite (10, 20, and 30 g/kg DM), and were evaluated *in vitro*.

#### 2.3.1. Gas Production Procedure

Method of the semi-automatic system of GP equipped with pressure transducer and a data logger (Pressure Press Data GN200, Sao Paulo, Brazil) as described by Bueno et al. [21] and adapted by Soltan et al. [22] was used to evaluate the dose–response effects of the experimental supplementations.

Rumen contents were collected freshly from adult fasted slaughtered of three Egyptian buffalo steers at the slaughterhouse of Faculty of Agriculture Alexandria University. The slaughtered animals were fed *ad libitum* a diet consisting of 50:50 commercial concentrate mixture: clover hay (*Trifolium alexandrinum* L.) and had free access to fresh water. Rumen contents were collected and kept separately in pre-warmed containers (39 °C) under anaerobic conditions. To prepare the rumen inocula (*n* = 3), the rumen content of each animal was blended for 10 s, squeezed through three layers of cheesecloth, and kept in a water bath (39 °C) under CO_2_ until inoculation took place. The different ruminal inocula were used to prevent the unusual effects of rumen environmental conditions [23,24].

Four analytical repetitions (4 bottles/inoculum/treatment) were used; two for the fermentation parameters and protozoal count, and the other two were for the determination of truly degraded organic matter (TDOM). Similarly, blank bottles (rumen fluid and buffer solution), and internal standard bottles (rumen inoculum, buffer solution, and clover hay) were prepared to correct for the sensitivity induced by the inocula [24,25].

Samples (0.5 g) of the experimental supplemented diets were weighed into numbered bottles and were incubated with 45 mL of diluted rumen fluid (15 mL mixed rumen fluid + 30 mL of Menkes buffered medium) in 120 mL incubation bottles [24,25]. Bottles were then sealed immediately with 20 mm butyl septum stoppers (Bellco Glass Inc., Vineland, NJ, USA), mixed, and incubated in a forced-air oven (FLAC STF-N 52 Lt, Treviglio, Italy) at 39 °C for 24 h. The gas head-space pressure of all bottles was recorded at 3, 6, 9, 12, 24 h incubation using a pressure transducer and a data logger (Pressure Press Data GN200, Piracicaba, Sao Paulo, Brazil). The pressure of GP in all bottles at each measuring time was converted into volumes to calculate the total accumulative gas produced through 24 h [22].

For CH_4_ determination through 24 h, one mL of gas of the bottle head-space was sampled by a syringe (med Dawliaico, Assiut, Egypt) at each gas pressure measuring time and accumulated in a 5 mL vacutainer tubes (BD Vacutainer^®^ Tubes, Franklin Lakes, NJ, USA). Methane concentration was determined using a gas chromatograph (Model 7890, Agilent Technologies, Inc., CO 80537, Santa Clara, CA, USA); the separation conditions were in detail described by Soltan et al. [22]. The amounts of CH_4_ produced were calculated according to Longo et al. [26]. Net values of both GP and CH_4_ were corrected for the corresponding blank values.

#### 2.3.2. Rumen Degradability 

At the end of the incubation, all bottles were put in cold water (4 °C) to stop the microbial fermentation process. Determination of TDOM was carried out according to Blümmel et al. [27] by immediate addition of neutral detergent solution (70 mL) without heat-stable α-amylase and incubated in a forced-air oven at 105 °C for 3 h. The remains were filtered in clean pre-weighed crucibles, washed with hot water, and dried at 105 °C for 16 h, and allowed to be burned at 550 °C for 4 h. The TDOM values were calculated from the difference between the amounts of the incubated OM and those remaining non-degraded. The portioning factor (PF) was calculated as the ratio of TDOM (mg) and gas volume (mL) [27].

#### 2.3.3. Rumen Fermentation Characteristics 

Rumen pH was determined using a pH meter (GLP 21 model; CRISON, Barcelona, Spain) in all fermentation bottles. Protozoal count was microscopically determined and differentiated by Digital Zoom Video microscope (LCD 3D, GiPPON; Wanchai, Hong Kong) following the procedure described by Dehority et al. [28]. 

Individual short-chain fatty acids (SCFAs) concentrations were determined according to Palmquist and Conrad [29] and adapted to Soltan et al. [22] using gas chromatography (Thermo fisher scientific, Inc., TRACE1300, Rodano, Milan, Italy) fitted with an AS3800 autosampler and equipped with a capillary column HP-FFAP (19091F-112; 0.320 mm o.d., 0.50 μm i.d., and 25 m length; J & W Agilent Technologies Inc., Palo Alto, CA, USA). A mixture of known concentrations of individual SCFAs was used as an external standard (Sigma Chemie GmbH, Steinheim, Germany) to calibrate the integrator. Concentrations of ruminal NH_3_-N were measured colorimetrically using a commercial lab kit (Biodiagnostic kits, Giza, Egypt) [30].

### 2.4. In Vivo Experiment

#### 2.4.1. Animals and Experimental Design

Based on the *in vitro* assay results, the most effective level of both natural and nano-zeolite was selected to evaluate their responses on apparent nutrients digestibility. Thirty female non-lactating Barki goats were randomly divided into three dietary treatments (*n* = 10/treatment) according to initial body weight (26 ± 0.9 kg SE bodyweight) as follows: control (the same control basal diet that was used in the *in vitro* experiment), natural zeolite (20 g/kg DM), and nano-zeolite (0.40 g/kg DM). Animals were fed their experimental diets *ad libitum.* Zeolite supplementation was orally administrated to ensure the complete dose was received.

Goats were fed twice daily at 08:00 and 16:00 and allowed free access to fresh water throughout the experimental period. Animals were adapted to the experimental diets for 15 days, followed by 7 days as a collection period. 

#### 2.4.2. Rumen Fermentation Parameters

Samples of rumen fluid (~30 mL) were collected using an esophageal probe 3 h after the morning feeding. The first 15 mL of the ruminal sample was discarded to avoid saliva contamination; all samples were then strained through three layers of cheesecloth and immediately subjected to ruminal pH using the same portable digital pH meter that was used in the *in vitro* assay. Ruminal individual SCFAs, total protozoa numbers, and NH_3_-N concentration were analyzed as described previously in the *in vitro* experiment. 

#### 2.4.3. Apparent Nutrients Digestibility

Fresh fecal samples (~40 g each) were obtained daily from each goat at 09:00 and 17:00, about 1 h post-feeding. Apparent nutrient digestibility was determined in which acid-insoluble fiber was used as an internal marker based on the relative concentrations of these nutrients in the feed and feces [20]. These samples were pooled per goat and stored at −20 °C for later analysis. At the end of this period, all the fecal samples were dried in a forced-air oven at 60 °C for 72 h, ground to pass through a 1 mm screen, and chemically analyzed for DM, OM, EE, NDF, and ADF as described previously. 

### 2.5. Statistical Analyses

All results were analyzed using the general linear model procedure (PROC GLM) procedure of SAS [31]. The *in vitro* gas production experiment was performed in one run for all treatments. The analytical replicates were averaged before statistical analysis, with each inoculum being the statistical replicate; thus, the statistical number of replications of treatments (*n* = 3) are the true replications. Orthogonal contrast statements were designed to test the linear and quadratic responses of each dependent variable to the increasing concentrations of nano or natural zeolite. The results of *in vivo* experiment were subjected to analysis of variance using the following statistical model as Yi = μ + Ti + ei, where Yi = observations mean, μ = overall mean, Tj = treatment effect, and ei = residual error. Differences between the treatments were considered significant at (*p* < 0.05), and trends were accepted if (*p* < 0.10). Tukey’s procedure for multiple comparisons was used to detect differences among means of the *in vivo* experiment.

## 3. Results

### 3.1. In Vitro Experiment

The effects of different levels of natural and nano-zeolite forms on ruminal GP, CH_4_, TDOM, and partitioning factors are presented in Table 3. The GP increased quadratically (*p* = 0.004) with increasing doses of nano-zeolite supplementations, while the natural zeolite did not affect the GP values. Linear reductions (*p* < 0.05) in CH_4_ production (related to the incubated DM and TDOM) consistent with tended increases (*p* = 0.09) in TDOM were observed by both zeolite form supplementations. The most significant CH_4_ reductions (49 and 15%) were achieved by supplementations of 20 g/kg DM natural zeolite, and 0.4 g/DM kg nano zeolite, respectively, compared to the control. Neither nano nor the natural form of zeolite supplementation affected the partitioning factor. 

The *in vitro* effects of natural and nano-zeolite forms on rumen protozoal count are presented in Table 4. Increases in the total protozoal count consistent with increases in *Diplodinium* sp. and *Epidinium* sp. were observed by nano zeolite (quadratic effect; *p* < 0.05) and natural zeolite (linear effect, *p* < 0.01) supplementations. Only natural zeolite supplementation increased linearly (*p =* 0.001) and quadratically (*p =* 0.02) the *Eudiplodinium* sp., while no effects were observed by nano zeolite treatments. Similarly, *Isotricha* sp. tended to be increased (linearly, *p* = 0.09, and quadratically *p* = 0.05) with the increasing levels of the natural zeolite, while neither *Entodinium* nor *Ophryscolex* sp. was affected by both zeolite supplementations. 

Quadratic increases (*p* < 0.05) in total SCFAs concentrations and acetate molar proportions by the natural zeolite, while no effects were observed by the nano zeolite form. Both zeolite forms linearly enhanced (*p* < 0.05) and tended to increase quadratically (*p* < 0.001) propionate to molar proportions. Ratio of C2:C3 declined linearly (*p* = 0.01) by nano-zeolite, and quadratic (*p* = 0.02) by natural zeolite supplementation. The ruminal pH was not affected by dietary levels of nano or natural zeolite, while only natural zeolite linearly decreased (*p* = 0.004) the NH_3_-N concentration (Table 5). 

### 3.2. In Vivo Experiment

The effects of zeolite type supplementation on ruminal fermentation characteristics and protozoal count are shown in Table 6. Nano-zeolite increased total SCFAs (*p* = 0.021) and butyrate (*p* = 0.001) concentrations compared to other treatments, while it decreased (*p* = 0.03) valeric molar proportion compared with the natural form of zeolite. Goats fed natural zeolite had an increase (*p* = 0.05) in ruminal pH compared with goats fed the control diet, while no differences were observed between both zeolite forms on ruminal pH.

Both natural and nano-zeolite forms declined (*p* < 0.001) NH_3_-N concentration compared with the control. Moreover, both nano and natural zeolite increased (*p* < 0.001) ruminal *Isotrica* sp. populations compared with the control, while no differences were detected among the experimental treatments on the other protozoal populations.

The digestibility coefficients of DM, OM, CP, and EE are shown in Table 7. Natural and nano zeolite supplemented diets did not affect the DMI and nutrients digestibility. 

## 4. Discussion

Both TEM and SEM images of the experimental nano zeolite indicated that the mechanical grinding of the natural zeolite reduced their particle size and successfully presented in the nano-scale. The zeta potential of nano zeolite was negative charges that favorably enhance the affinity of palygorskite with cationic matters, e.g., cationic dyes, and then enhance the adsorption capacity. Results of the FTIR of the nano form of zeolite indicated the high efficiency of the performed nano-particles; 20 well-defined peaks of zeolite functional groups were observed, and 12 of them were found in higher frequencies at 2126–4009, while only four peaks appeared in the lower frequency range (from 466 to 789). These physico-chemical properties of the zeolite nano-form may result in different effects in rumen fermentation compared to its natural form. Increases in GP values caused by nano-zeolite addition may indicate the higher efficiency of nano-zeolite to improve the ruminal microbial fermentation than the natural zeolite; this can be due to the large surface areas, large capacity for cation exchange, and high activities caused by the size–quantization effect [32]. Rumen CH_4_ production is strongly related to microbial fermentation extent; therefore, enhancements in GP and nutrient degradability can increase rumen CH_4_ emission [33]. Thus, such increases in the total GP caused by the nano zeolite may partly explain the low efficiency of the nano-zeolite to reduce CH_4_ production compared to the normal form. Reductions in CH_4_ production caused by nano or natural forms confirmed the anti-methanogenic activity of zeolite in this study. Zeolite may act as an alkalinizer and has a high capacity for H+ exchange at different pH ranges [34,35]. Therefore, zeolite can reduce CH_4_ emission by affecting rumen H+ exchange capacity and can also affect all the end fermentation characteristics. 

Most common CH_4_ inhibitors may adversely affect the ruminal nutrient degradability and/or microbial fermentation at doses that achieve desirable CH_4_ reduction [22]. In the *in vitro* study, CH_4_ reduction consistent with increases in TDOM and GP caused by nano zeolite supplementations may indicate that both zeolite types might benefit the alteration of ruminal fermentation pattern towards less CH_4_ production without adverse effect on feed degradability. This can be due to the catalytic activity of zeolite nano-particles which can increase some digestive fiber enzymes (such as amylase, α-amylase) to improve OM degradability [36]. Moreover, the literature reported that enhancing the rumen nutrient degradability is a typical action of zeolite through the buffering effect and maintaining the ruminal pH from the rapid decrease [36]. In the current study, both zeolite forms enhanced the TDOM *in vitro* and the pH *in vivo*, while no differences were detected in the nutrient digestibility *in vivo*. The reasons for this phenomenon are not clear, but it seems that the activity of zeolite is more efficient in the rumen than in the post-ruminal digestive tract parts. The current results are in line with Galindo et al. [37], who reported that zeolite could provide favorable conditions for the increase of cellulolytic rumen bacteria and subsequently increase the ruminal degradable organic matter. The lacking effects of zeolite supplementation on apparent nutrient digestibility are consistent with what reported by Câmara et al. [38], as total tract DM digestion was unaffected when zeolite supplemented at levels of 30–50 g zeolite/kg of dietary DM, while McCollum et al. [39] observed enhancements in ruminal digestion of OM and starch with supplementation of 25 g zeolite/kg of finishing diet.

Reduction in CH_4_ can be achieved indirectly by decreasing protozoal abundance [22], but results of the current study indicated that CH_4_ reduction was consistent with increases in the total protozoal count, which is mainly related to the significant increases in *Diplodinium* sp., the highest number of protozoal species naturally found in the typical rumen conditions of ruminants. This may partly explain the high TDOM caused by both zeolite types, as *Diplodinium* sp. is known for the high efficiency for cellulose degradation, and consequently, H+ abundance [33,34,35,36,37,38,39,40]. Therefore, the current results indicated that the CH_4_ reduction was a result of the high rumen H+ exchange capacity of zeolite.

According to some previous studies [40,41], ruminal pH stability provides more favorable environmental conditions for more microbial proliferation. In the present study, the observed increase in total protozoal abundance may be due to the practical stability of rumen pH within the normal range associated with the available energy (as SCFAs production) and nitrogen (as adequate NH**_3_**-N concentration) for more microbial protein synthesis. This explanation agrees with Dschaak et al. [42], who reported that the great affinity of zeolites for holding water and osmotically active cations could enhance ruminal microbial fermentation and osmotic activity that can regulate pH in the rumen by buffering against hydrogen ions of organic acids. Our results also confirmed that rumen pH plays an important role in the survival of rumen -ciliated protozoa [40,41].

The SCFAs patterns of both zeolite forms declared the ability of nano zeolite to modify the microbial fermentation activity differently from its natural form. The *in vitro* experiment revealed that natural zeolite quadratically enhanced acetate concentration; consequently, the total SCFAs (as acetate is the main contributor of total SCFAs), while these were not caused by the nano form of zeolite. Additionally, the nano form of zeolite enhanced butyric concentration in the *in vivo* experiment compared with the natural zeolite form. These differences may confirm our suggested hypothesis that performing the nano form of the zeolite may affect their efficiency as feed additive differently from its natural form. 

Results of the *in vitro* assay showed that both zeolite forms enhanced propionate molar proportions concentration. These results, alongside decreases in acetate to a propionate ratio, might be due to shifting SCFAs production pattern from acetate toward more propionate production, which may explain that the fermentation process occurred in a more efficient manner where more hydrogen ion (H**^+^**) may be used by ruminal microbes to synthesize SCFAs (propionate) rather than CH**_4_**. 

Additionally, differences in fermentation patterns were observed by the *in vivo* and *in vitro* experiments using the same experimental dose. It seems that the time of collection of the ruminal samples (3 h post-feeding) of the *in vivo* assay, rather than the nutritive buffering solution used in the *in vitro* assays, may affect the obtained results.

Both zeolite forms decreased NH_3_-N concentration in the *in vivo* assay, while it occurred only by the natural form in the *in vitro* experiment. Zeolite, as a cation exchanger, is capable of exchanging and holding the ammonium ion before its release by the sodium ion (Na^+^) present in the saliva that was entering the rumen [43,44,45]. In this regard, zeolite additive could exhibit a higher potential to sink hydrogen through its cation exchange capacity, which might be another possible explanation for zeolite-buffering properties. Lower ruminal NH**_3_**-N concentration with the addition of nano-zeolite indicated that zeolite was able to capture NH**_3_** through the character of cation exchange capacity [46].

The modulation of rumen fermentation patterns that occurred by both zeolite forms may be nutritionally advantageous for lactating and growing ruminants through enhancing ruminal OM degradability and propionate production [47].

## 5. Conclusions

The nano transformation of the natural zeolite positively affected the physico-chemical properties of the natural zeolite. Zeolite, whether in its natural or nano-form, was able to maintain rumen pH while reducing NH3-N concentration and affecting CH4 production without adverse effects on the apparent nutrient digestibility. Zeolites as clay minerals play a role in improving the rumen environment and fermentation end-products because of their buffering role. In both experiments, nano zeolite modified the SCFAs pattern differently from the natural zeolite. These results may suggest that the consideration of zeolite as a modifier of rumen fermentation was not only dose-dependent but also particle-size-dependent.

## Figures and Tables

**Figure 1 animals-11-02215-f001:**
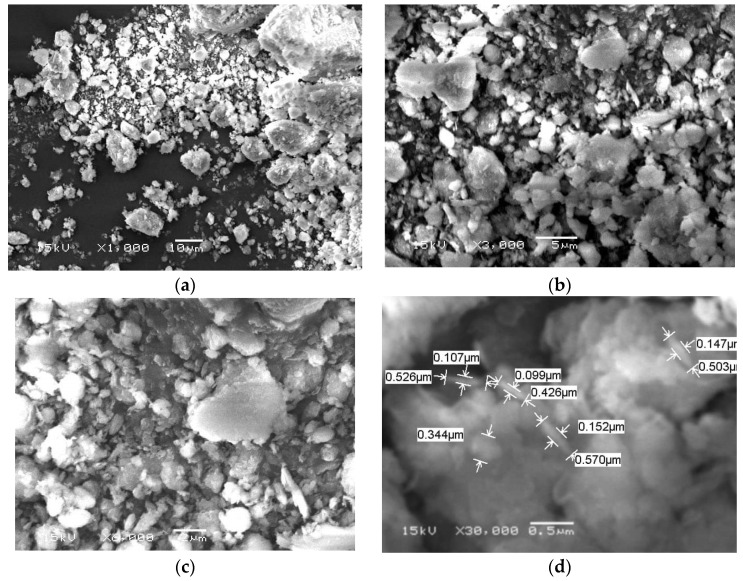
The surface morphology of the nano-zeolite by scanning electron microscope (SEM): (**a**) SEM with X1000; (**b**) SEM with X3000; (**c**) SEM with X6000; (**d**) SEM with X30000.

**Figure 2 animals-11-02215-f002:**
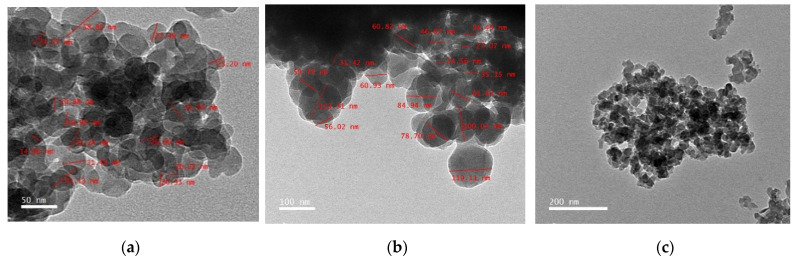
The nano-particles size and shape of the nano-zeolite by transmission electron microscope (TEM): (**a**) TEM with 50 nm; (**b**) TEM with 100 nm; (**c**) TEM with 200 nm.

**Figure 3 animals-11-02215-f003:**
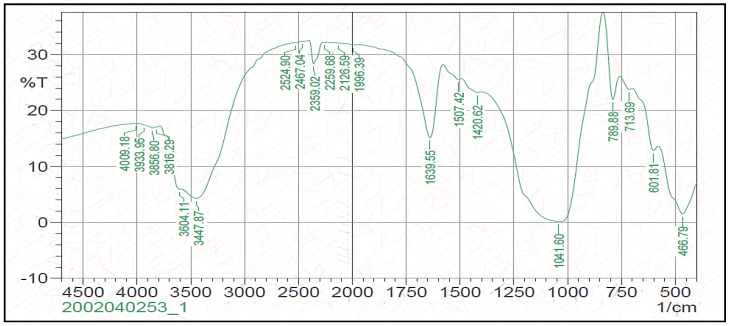
Fourier transform infrared spectroscopy (FTIR) spectra for the experimental nano-zeolite.

**Figure 4 animals-11-02215-f004:**
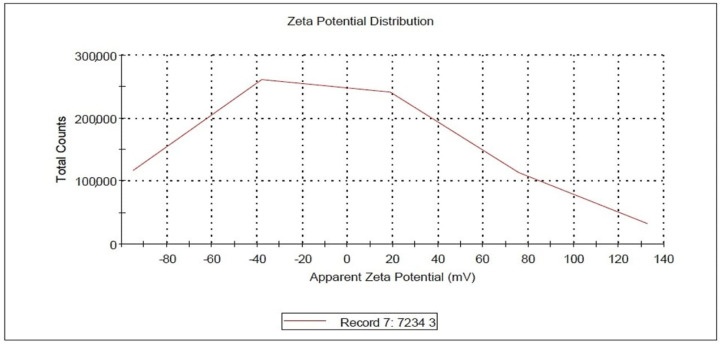
Zeta potential distribution of the experimental nano-zeolite.

**Table 1 animals-11-02215-t001:** Chemical composition and physical properties of the natural form of zeolite.

Item	Zeolite Characteristics
Chemical composition	
SiO_2_	650.0–713.0 g/kg
Al_2_O_3_	115.0–131.0 g/kg
CaO	27.0–52.0 g/kg
K_2_O	22.0–34.0 g/kg
Fe_2_O_3_	7.00–19.0 g/kg
MgO	6.00–12.0 g/kg
Na_2_O	2.00–13.0 g/kg
TiO_2_	1.00–3.00 g/kg
Si/Al ratio	4.80–5.40
Physical properties	
Softing point	1260 °C
Melting point	1340 °C
Flow temperature	1420 °C
Specific gravity	2200–2440 kg/m^3^
Volume density	1600–1800 kg/m^3^
Porosity	24–32%
Compactness	70%
Whitens	70%
Appearance	Gray-green

**Table 2 animals-11-02215-t002:** Ingredients and chemical composition of experimental basal diet used in the *in vitro* and *in vivo* experiments.

Item	Basal Diet (g/kg Dry Matter)
Ingredients	
Berseem clover hay	500
Ground yellow corn	345
Soybean meal	150
Mineral and vitamin mixture ^1^	5.00
Chemical composition (g/kg DM)	
Organic matter	924
Crude protein	131
Ether extract	20.0
Neutral detergent fiber	718
Acid detergent fiber	343
Acid detergent lignin	60
Hemicellulose	375
Cellulose	283

^1^ Each kg contained: 45.8 g dicalcium phosphate, 15 g magnesium sulfate, 6.15 g ferrous sulfate, 0.393 g potassium iodide, 0.753 g copper sulfate, 0.248 g cobalt sulfate, 0.373 g zinc sulfate, 0.02 g slinat sodium.

**Table 3 animals-11-02215-t003:** Supplementation effects of natural and nano-zeolite forms on ruminal gas production (GP), methane, truly degraded organic matter (TDOM), and partitioning factor through 24 h incubation period (*in vitro* experiment).

Treatment	GP (mL/g DM Incubated)	Methane		TDOM (g/kg)	Partitioning Factor (mg TDOM/mL GP)
		(mL/g DM Incubated)	(mL/g TDOM)		
Control	133	7.7	10.8	709	1.10
Nano zeolite (g/kg DM)					
0.20	143	10.51	14.33	730	1.11
0.30	129	10.12	13.80	733	1.13
0.40	142	6.74	9.13	756	1.18
0.50	153	6.94	9.90	734	1.09
1.00	141	7.32	10.2	742	0.97
Contrast 1					
SEM	1.40	0.22	0.38	8.19	0.01
Linear	0.36	0.01	0.05	0.09	0.56
Quadratic	0.004	0.04	0.10	0.46	0.86
Natural zeolite (g/kg DM)					
10	138	4.99	7.40	710	1.15
20	137	3.98	5.50	741	1.13
30	140	5.15	7.25	711	1.16
Contrast 2					
SEM	0.42	0.06	0.11	2.45	0.002
Linear	0.25	0.002	0.001	0.09	0.62
Quadratic	0.36	0.16	0.24	0.36	0.50

Contrast 1 = effects of control (0 supplementation g/kg DM) compared with nano zeolite supplementations, and Contrast 2 = effects of control (0 supplementation g/kg DM) compared with natural zeolite supplementations. SEM: standard error of the mean.

**Table 4 animals-11-02215-t004:** Supplementation effects of natural and nano-zeolite forms on ruminal protozoal count through 24 h incubation period (*in vitro* experiment).

Treatment	Protozoal Count (×10^5^/mL)						
	*Diplodinium*	*Entodinium*	*Epidinium*	*Eudiplodinium*	*Isotricha*	*Ophryscolex*	Total
Control	8.49	1.42	0.412	0.150	0.26	0.150	10.9
Nano zeolite (g/kg DM)							
0.20	10.4	1.20	0.07	0.11	0.15	0.22	12.1
0.30	8.62	1.12	0.15	0.37	0.15	0.11	10.5
0.40	12.9	1.53	0.07	0.30	0.41	0.15	15.4
0.50	10.0	1.46	0.01	0.67	0.30	0.07	12.5
1.00	11.5	1.39	0.03	0.90	0.41	0.11	14.3
Contrast 1							
SEM	0.39	0.143	0.045	0.082	0.098	0.052	0.44
Linear	0.86	0.28	0.01	0.18	0.49	0.71	0.67
Quadratic	0.009	0.75	0.03	0.31	0.68	0.28	0.05
Natural zeolite (g/kg DM)							
10	11.7	1.20	0.07	0.11	0.94	0.11	14.2
20	13.5	1.01	0.11	0.83	0.71	0.30	16.5
30	11.6	1.20	0.04	0.22	0.37	0.26	13.7
Contrast 2							
SEM	0.124	0.043	0.013	0.024	0.029	0.015	0.1334
Linear	<0.001	0.20	0.04	0.001	0.09	0.21	<0.001
Quadratic	0.36	0.95	0.13	0.03	0.05	0.27	0.61

Contrast 1 = effects of control (0 supplementation g/kg DM) compared with nano zeolite supplementations, and Contrast 2 = effects of control (0 supplementation g/kg DM) compared with natural zeolite supplementations. SEM: standard error of the mean.

**Table 5 animals-11-02215-t005:** Supplementation effects of natural or nano-zeolite on ruminal total short-chain fatty acids (SCFAs) concentration, molar proportions of individual SCFAs, pH, and ammonia nitrogen (NH_3_-N) concentrations through 24 h incubation period (*in vitro* experiment).

Item	SCFAs (% of Total SCFAs)							Total SCFAs (mM)	pH	NH_3_-N (mg/100 mL)
	Acetate	Propionate	Butyrate	Iso-butyrate	Valerate	Iso-Valerate	C2:C3			
Control	59.1	22.2	11.6	0.297	2.02	3.31	2.66	90.0	6.34	20.7
Nano zeolite (g/kg DM)										
0.20	58.4	24.9	11.1	0.276	1.96	3.30	2.34	103	6.31	21.1
0.30	58.7	24.8	10.9	0.282	1.95	3.37	2.37	99.0	6.32	19.3
0.40	58.6	24.4	11.3	0.309	1.97	3.26	2.36	94.0	6.26	16.3
0.50	60.3	24.7	10.2	0.209	1.71	2.85	2.44	95.0	6.25	19.8
1.00	59.7	23.1	11.4	0.310	1.99	3.39	2.59	89.0	6.32	20.6
Contrast 1										
SEM	0.631	0.316	0.622	0.033	0.074	0.146	0.041	3.150	0.012	0.475
Linear	0.71	0.007	0.56	0.82	0.68	0.87	0.01	0.29	0.54	0.27
Quadratic	0.65	0.06	0.86	0.82	0.88	0.91	0.07	0.24	0.49	0.33
Natural zeolite (g/kg DM)										
10	63.7	21.6	9.88	0.18	1.61	3.03	2.82	83.0	6.36	20.3
20	59.4	24.8	10.7	0.21	1.75	2.94	2.39	100	6.33	17.9
30	58.2	23.4	12.6	0.32	1.95	3.40	2.49	103	6.34	19.2
Contrast 2										
SEM	0.19	0.09	0.186	0.010	0.022	0.044	0.012	0.94	0.004	0.14
Linear	0.86	0.001	0.64	0.43	0.16	0.47	0.10	0.14	0.83	0.004
Quadratic	0.03	0.08	0.421	0.44	0.11	0.82	0.02	0.05	0.15	0.13

Contrast 1 = effects of control (0 supplementation g/kg DM) compared with nano zeolite supplementations, and Contrast 2 = effects of control (0 supplementation g/kg DM) compared with natural zeolite supplementations. C2:C3 = acetate to propionate ratio. SEM: standard error of the mean.

**Table 6 animals-11-02215-t006:** Supplementation effects of natural or nano-zeolite on goat rumen fermentation parameters and protozoal count 3 h post-feeding (*in vivo* experiment).

Items	Treatments			SEM	*p*-Value
	Control	Natural Zeolite	Nano Zeolite		
Total SCFAs, mM	72.3 ^b^	74.3 ^b^	86.8 ^a^	2.48	0.02
SCFAs (% of total SCFAs)					
Acetic	64.3	65.8	61.6	0.93	0.20
Propionic	17.1	15.7	16.0	0.56	0.50
Isobutyric	2.13	2.30	1.93	0.10	0.41
Butyric	9.92 ^b^	10.2 ^b^	14.3 ^a^	0.66	0.001
Isovaleric	4.37	3.68	4.45	0.42	0.76
Valeric	2.24 ^ab^	2.46 ^a^	1.74 ^b^	0.12	0.03
C2:C3	3.83	4.25	3.95	0.19	0.69
pH	5.50 ^b^	5.96 ^a^	5.72 ^ab^	0.07	0.005
NH_3_-N, (mg/100 mL)	6.28 ^a^	5.16 ^b^	3.90 ^c^	0.30	0.001>
Protozoa, (×10^5^/mL)					
* Diplodinium*	10.33	12.3	11.4	0.39	0.113
* Entodinium*	1.37	1.50	1.11	0.09	0.212
* Epidinium*	1.07	1.13	1.17	0.11	0.932
* Eudiplodinium*	0.57	0.70	0.63	0.08	0.793
* Isotrica*	0.23 ^c^	0.77 ^a^	0.50 ^b^	0.06	<0.001
* Ophryscolex*	0.17	0.13	0.40	0.06	0.123
Total	13.7	16.8	15.3	0.65	0.147

^a,b,c^ Means within a row without a common superscript letter differ significantly at *p* < 0.05. SCFAs = short chain fatty acids concentration. % of total SCFAs = molar proportions of individual SCFAs. NH_3_-N = ammonia. C2/C3 =acetate to propionate ratio. SEM= standard error of the mean.

**Table 7 animals-11-02215-t007:** Supplementation effects of natural or nano-zeolite on dry matter intake (DMI) and apparent nutrients digestibility of goats (*in vivo* experiment).

Items	Treatments			SEM	*p*-Value
	Control	Natural Zeolite	Nano Zeolite		
DMI (g/day)	1167	1184	1179	13.21	0.904
Digestibility (g/kg)					
Dry matter	443	446	445	0.13	0.56
Organic matter	435	459	445	0.61	0.31
Ether extract	556	565	608	1.42	0.30
Crude protein	387	386	425	0.90	0.12
Neutral detergent fiber	429	436	425	0.43	0.62
Acid detergent fiber	355	362	341	0.66	0.45
Hemicellulose	505	513	513	0.44	0.764
Cellulose	472	467	447	0.75	0.389

## Data Availability

Not applicable.

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
