# Peer review of "In Vitro and In Vivo Assessment of Dietary Supplementation of Both Natural or Nano-Zeolite in Goat Diets: Effects on Ruminal Fermentation and Nutrients Digestibility"

_animals, 2021, doi:10.3390/ani11082215_

Round 1
Reviewer 1 Report
This manuscript tries to assess the effect zeolites and nanozeolites on in vitro rumen fermentation parameters and apparent digestibility. I find the manuscript subject interesting for the audience of Animals but some points need to be addressed before I can consider this manuscript acceptable for publication. Therefore, my decision is to reconsider after major revision.
Major concerns
- The introduction needs to be improved by clearly stating the hypothesis of the manuscript. It is missing as it is right now. In addition, authors need clearly state the novelty of the current manuscript.
- A very common design is to examine incremental additions of a nutrient, or compound, to a diet, such is the case of the current trial, in order to identify an optimal level and/or response surface. In such cases, multiple comparison tests are unlikely to be appropriate and polynomial contrasts are generally the most effective statistical test to identify the form of a response to an increasing level of a treatment. Therefore, my suggestion is to delete the multiple comparison test in the in vitro trial and describe results just according to the linear and/or polynomial effects.
- The results, discussion and conclusion sections should be reworded according to the new statistical analysis.
Minor comments
Line 1-2. I suggest mentioning the natural zeolite as well. Otherwise it is not clear why authors use it as well.
Line 55-56. Very arguable and it is in contradiction with Lines 59-60. Please reword
Line 76-77. I suggest spending a few more sentences to clearly highlight the novelty of the current trial. In my opinion it is not enough with these two lines.
Line 78-81. I suggest rewording the objective. If this is the objective why using so many different doses? would it not be to determine the response with increasing doses or try to assess the optimal dose?
Table 1. Are these theoretical values or determined? If determined these information is missing in the material and methods section.
Line 107-108"..which are very useful..." These is discussion. Either delete or reword.
Line 140-142. This explanation should be better given in the discussion section.
Table 2. Are these values calculated or determined? if determined information on these chemical analyses is missing in the material and methods section.
Line 182-183. Confusing. Please reword.
Line 227 colorimetrically instead of calorimetrically?
Line 232-233. Were animals individually or group fed? If intake was measured to estimate apparent digestibility I suggest reporting intake in Table 6.
Line 242. Description on how AIA was analysed is missing
Line 244. bulked by animal? it is important to make this clear so that no more than 30 determinations (experimental unit) were used in the statistical analyses.
Table 2-7. Use of decimals. Use 3 decimals with means below 0, 2 decimals with means between 1 and 10, 1 decimal with means between 11 and 100, and do not use decimals with means greater tan 100. Provide SEM values with one decimal more than that used for the lsmean. Provide P values rounded to the third decimal.
Table 3. Methane expressed as g TDOM. Please check the heading. It is confusing.
Table 5 and 7. Report the branched-chain VFA and the (C2+C4)/C3 ratio
Line 315-317. Only report significant differences
Line 328-330. This is discussion. Please report differences compared to control.
Table 7. Report units
Author Response
Dear Prof. Reviewer (1)
We would lie to appreciate you so much for your valuable comments on the Manuscript ID: animals-1216706, which we have taken at our considerations.
- Regarding to your comments about the introduction section to clearly state the hypothesis and novelty of the manuscript.
- We have done at the manuscript
- Regarding to your comments about the statistical analysis of the data
- The data were reanalyzed and the orthogonal contrast statements were designed to test the linear and quadratic responses of each dependent variable to the increasing concentrations of nano or natural zeolite.
- The results were reworded according to the new statistical analysis.
- Minor comments
- We have done at the revised manuscript
We wish that our responses to your comments will be satisfied
Regards and thanks again

Reviewer 2 Report
Dear Authors,
The manuscript titled “In vitro and in vivo assessment of dietary supplementation of nano-zeolite in goat diets: Effects on ruminal fermentation and nutrients digestibility” has been reviewed.
Thank you for this opportunity you have given me to review this manuscript, which aimed to evaluate in vitro and in vivo dietary supplementation with different levels of natural or nano zeolite forms on rumen fermentation patterns and nutrient digestibility.
Although it seems that authors have done lots of works to generate data, unfortunately, the manuscript cannot be accepted in its present form.
Generally, authors should follow the guidelines of writing given by the journal. For instance, the subtitles, table captions, and reference citations are written incorrectly
Major comments:
Simple Summary
Lines 17-29: As mentioned in the Journal guidelines, simple summary should be written in layman’s term for your audience to grasp what you want to show them; however, I found it similar to your abstract. I suggest to limit the highly technical terms and make it simple and concise to benefit your lay readers. Also, avoid using abbreviations.
Line 17: Change “me-thane” to “methane”.
Line 21: “Natural” should be in lowercase.
Abstract
Lines 30-48: Limit your abstract to about 200 words maximum as stated in the guidelines.
Line 29,30,34: Acronyms should be spelled-out in the first appearance in abstract, main text and tables/figures, and then abbreviation may use thereafter.
Introduction
Line 49: “Akin”. I suggest to use other word.
Lines 72,75, and elsewhere in the Discussion part: check the citation. Please revise according to the format.
Line 79: Remove the other “of” in the sentence and be consistent in using the word “nano-zeolite”.
Materials and Methods
- Line 90: remove the colon “:” in section 2.1.
- Lines 92-93: I suggest better to include this description in the Introduction part rather in Materials and Methods.
- Line 95: change “table” to “Table”.
- Line 97: Bold the “Table 1”, remove the colon (:) and change it into a period (.) instead. Put also a period (.) after the table caption.
- Line 98-99: Check the table format. What does asterisk (*) means in Table 1? What are those numbers in the opposite of the zeolite composition? I think “Zeolite characteristics” is not suitable to pertain those numbers. Remove the line in “g/kg”, “Physical properties”, and in “º”. Remove the space in between “º” and “C” and it should be “°C”.
- Lines 100-103: Needs to reconstruct the sentence. It’s kind of confusing. Put the product company in proper place. In line 101, put a space in between “6” and “h”. The symbol “)” in line 103 made me confused. Remove the line in “United States” in line 104. Please confirm if the “rang” in line 105 is supposedly “range”.
- Line 108: Please remove the “(SEM) and (TEM)” and put them instead in the proper location. For instance, (SEM) should be put in line 106 right after the “scanning electron microscope”; and (TEM) in line 113 after the “transmission electron microscope”. Better change “showed” to “shown”.
- Line 110: Is “Jeol JSM-6360 LA” really a machine name or the manufacturer? Please confirm.
- Line 112-113: refer to comment in line 108 about the TEM.
- Line 114: You can now use here the acronym “TEM” to denote the transmission electron microscope.
- Lines 118-120: Remove the colon (:) after the “Figure 1” and replace with a period (.) instead. Be consistent in using “nano-zeolite”. There should only be one format for the entire manuscript. In the title, authors used “nano-zolite”, therefore, it should be maintained and consistent in all parts of the manuscript. In line 119, the figure caption shows “(c)” to denote SEM with X6000; however, in the figure description the authors used “(C)”. Please carefully check.
- Lines 122-123: The same comment as with Lines 118-120. Please check my comments. In addition, put a space in between number and unit. For example, (a) TEM with 50 nm. Change the “100nm” to “100 nm” etc..
- Line 124: Change “nano zeolite” to “nano-zeolite”. Please check and change them all.
- Line 127: Remove the parenthesis in “Figure (3)”.
- Lines 129-133: Please check the sentence structure. It is kind of confusing. In line 132, put a space in between “and” and 0.00165. Also add a comma (,) right after “respectively”. In line 133, remove the parenthesis in “(Figure 4)”.
- Line 135: Replace the “:” with “.” in “Figure 3:”. Check my comment in “Nano zeolite”.
- Line 137: Replace the “:” with “.” in “Figure 4:”. Change “zeta” to “Zeta” since it is the first word of the sentence. Check my comment in “Nano zeolite”.
- Line 138: remove the colon “:” in section 2.2.
- Line 141: Check my comment in “nano zeolite”. Be consistent.
- Line 143: Change the “roughage: concentrate” to “roughage:concentrate”. No space in between “:” and “concentrate”.
- Line 144: You need to show the reference name instead of using the citation number “[18]”. Please check the proper citation in the journal guidelines.
- Lines 147-148: Bold the “Table 2”, remove the colon (:) and change it into a period (.) instead. Put also a period (.) after the table caption. Check the table format.
- Lines 162-165: Please check and use the journal table format. It is kind of confusing. Check also the proper format of reference citation.
- Lines 170-173: Please check the proper use of reference citation.
- Lines 174-183: This section is questionable because the study focused on goat diet; however, in this part of methodology, rumen contents were collected from the slaughtered buffalo steers. You cannot draw any conclusion on the effect of treatments in goat diet since you used the wrong animal for this particular experiment. Another thing, are authors sure of the animal diet used for rumen content sampling? Does 500 g commercial concentrate and 500 g clover enough for the buffalo? In line 180, you have stated rumen inocula, what do you mean here?
- Line 184: Is inoculum really the right term you used here? Please check.
- Line 190: change “weighted” to “weighed”
- Line 195: change “39o C” to “39 °C”
- Line 196: provide space in between “24” and “h”
- Lines 202, 203, 208: check the proper format of citation.
- Line 205: “rumen” is not in italic form. Remove the colon (:)
- Lines 206,209,210,211: refer to comment in line 195.
- Line 215: remove the colon (:)
- Lines 219,221: check the proper format of citation
- Line 228: check the format
- Lines 233,234,238: check my comments in “nano-zeolite”
- Line 236: I couldn’t fine peanuts hay in Table 2. Please confirm.
- Line 245: check the format
- Line 246: put space between “3” and “h”.
- Line 256: In the statistical model you have used, what does T1+ means? However in the derivation, you defined Tj as treatment effect. Therefore, which is Tj in the formula?
Results
- Line 263: The subtitle 3.1. should be italicized and remove the colon (:) at the end.
- Line 264 and other lines: Check my comment in “nano zeolite”. Is it hyphenated or not? Be consistent.
- Line 266 and other lines: Based on the journal format, the “p value” should be written this way (example: p < 0.05; p = 000). The “p” is in italic form followed by non-italicized characters. Therefore, your result should be written this way (p = 0.001) instead of (P=0.0005). You also need to convert it in 3 decimal places. Please convert all SEM and p-values in your result to maintain the consistency and for better presentation of your data.
- Lines 268: “natural zeolite”. Using this terminology is kind of confusing. The items you are presenting in your tables are “nano-zeolite” and “normal zeolite”. Be consistent in your terminology.
- Lines 269,272, etc.: Check my comment in “nano zeolite”. Be consistent.
- Line 272: “/kg” is better change to “per kg”
- Line 274: Based on the journal format, the “p value” should be written this way (example: p < 0.05; p = 000). Check all and change as suggested.
- Line 276: What do you mean by “Natural”? In Table 3, there are just 2 treatments which include Nano zeolite and Normal zeolite. Be consistent.
- Lines 277-279: Results of TDOM and PF were not significant but there were superscripts denoting for significant effects. Please interpret clearly. Another, “/kg” is better change to “per kg”.
- Lines 280-283: Check the table format, for instance, “Table 3:” should be “Table 3.” and put a period (.) at the end of the table caption. Also change the table style based on journal format. You can check published manuscript as reference. Check the superscript in the table footnote denoting the significance (a,b,c). As shown in your table, superscript is until (d).
- Line 284: The subtitle 3.2. should be italicized and remove the colon (:) at the end.
- Line 285: check my comment about “nano zeolite”.
- Line 286: “Tables” should be “Table”. Remove the “s” from it. “abundant” should be “abundance”. Check my comment in “p values”, for instance you “P < 0.001”.
- Line 287: change “/kg” to “per kg”. As shown in Table 4 under normal zeolite, different concentrations are statistically comparable with each other; however, you only stated the high abundance at 10 and 20 g/kg DM of normal zeolite. Also, be consistent in your terminologies, for instance, you used “normal zeolite” in your table while in the result part you used “natural zeolite”. This will cause confusions to your readers.
- Line 288: check my comment in p “spp” should not be in italic form and better change it to “sp.”. The statement is misleading. It shows in the table that treatments did not affect significantly the Entodinium; however, authors are stating in the result the treatments’ significant effect on this species of protozoa. In Table 4, p value under Entodinium is 0.711 but in your result interpretation, you are stating similar trend (p < 0.001) which is contradicting with your presented data, thus made me confused.
- Line 290: check my comment in “natural zeolite”, “p value”, and “spp”.
- Lines 292-295: check my comment in “nano zeolite”, “/kg”, and “sp”. In lines 294-295, it is kind of confusing. Treatments significantly affects the Diplodinium based on your data in Table 4. Result on Isotricha is not presented.
- Line 296: Check comment in line 284.
- Line 297-300: check my comment in p
- Line 299: include also the result of “control” in your statement.
- Line 300: natural or normal? Please be consistent.
- Line 302: No pH data in Table 3. Check very carefully.
- Line 305: Check my comment in line 280 about the table format.
- Lines 305-307: Scientific names should be italicized. Remove space in between “/” and “ml” in the Protozoal count unit in the table. Superscript indicating mean for the significant differences should be until “e”.
- Lines 308-309: Check my comment in previous lines about the table format including the caption.
- Lines 308-310: better to change “C2:C3” to “A:P” which is commonly used in all manuscripts. “NH3-N” to “NH3-N”.
- Line 312: Check the format.
- Line 313: remove parenthesis in “(6)”.
- Lines 318-319: Check the table caption format.
- Lines 318-323: Be consistent in using terminologies. “Natural zeolite”, are you pertaining to normal zeolite? If so, then use the same terminology throughout the manuscript.
- Lines 320-323: Minimize the use of superscript in the footnote. Define the acronym in simplest way without using superscript. Refer to published manuscript.
- Line 325: Check the format.
- Line 327: “table (7)” should be “Table 7”. Check the format of p values.
- Line 330: “ammonia” should be “ammonia nitrogen”.
- Line 331: “spp” is not in italic form.
- Line 333-335: Check the table format. “Natural zeolite” to “normal zeolite”. Check the comment in format of p values. A:P is better to use than C2:C3. What does the “+3h” means in this table? “spp” should not be in italic form.
Discussion
I did not go further in this part since I found many errors which affects the quality of the manuscript. It seems that authors did not properly check the guidelines of the journal creating inaccuracy and affects the harmony of the manuscript.
Author Response
Dear Dear Prof. Reviewer (2)
We would lie to appreciate you so much for your valuable comments on the Manuscript ID: animals-1216706, which we have taken at our considerations.
- Lines 174-183: This section is questionable because the study focused on goat diet; however, in this part of methodology, rumen contents were collected from the slaughtered buffalo steers. You cannot draw any conclusion on the effect of treatments in goat diet since you used the wrong animal for this particular experiment.
- The slaughtered buffalo steers were considered as rumen content or inolcula donors only for gas production assay in vitro
- Another thing, are authors sure of the animal diet used for rumen content sampling?
- These animals were fed and slaughtered at the slaughter house of the faculty of Agriculture, Alexandria University
- Does 500 g commercial concentrate and 500 g clover enough for the buffalo?
- This is not feeding level and this mean that roughage: concentrate ratio was 50:50 or (1:1) and of course is not enough for feeding buffalo steers. This sentence was reworded
- In line 180, you have stated rumen inocula, what do you mean here?
- Rumen inocula is rumen content (sloid and liquid) for the in vitro assay of gas production
- Minor comments
- We have done at the revised manuscript
We wish that our responses to your comments will be satisfied
Regards and thanks again

Reviewer 3 Report
Line 17: switch me-thane to methane
Line 39: delete the space in "g/ kg DM" to read "g/kg DM"
Line 61: long sentence, please split into two.
Line 79: Delete the word "of", as it is used two times.
Table 1: For the Zeolite composition column you list the units as g/kg and then %, but then you list the units of the physical properties in the second column. please keep all measurement units in the same column. Also, where is the key for the astric used after "Zeolite characteristics*"?
line 184: delete the space before the word treatment
Lines 209, 211, 244 and others: please check how you reported temperature throughout the study and fix accordingly so that they are consistent throughout the paper.
Lines 277, 278, and throughout the paper: please fix how you report your P-values. They should all be a capitalized P that is italicized. check throughout the entire paper.
P-values reported in table 3 should be consistent with a 0 added before the decimal. Fix the P-values under the column for methane.
There are many spacing inconsistencies throughout the entire paper next to P-values, = signs, or units of measurement. the entire paper should be read over to address these issues.
In the conclusion, authors report that both natural and nano-form zeolite improve propionate production. However, results and P-values from Table 7 indicate that there were no differences from control.
Author Response
Dear Prof. Reviewer (3)
We would lie to appreciate you so much for your valuable comments on the Manuscript ID: animals-1216706, which we have taken at our considerations.
- Minor comments
- We have done at the attached revised manuscript
We wish that our responses to your comments will be satisfied
Regards and thanks again

Round 2
Reviewer 1 Report
Authors have followed most of my suggestions but some more need to be addressed before I can consider this manuscript acceptable for publication. Therefore, my decision is to reconsider after major revision.
Major comments
The presentation of results needs to be improved. When describing a quadratic impact in the text, it is critical that its form be defined. For example, only a significant quadratic effect could lead to a statement such as: ‘values were highest/lowest at the intermediate addition level’, whereas significant linear and quadratic effects could lead to a statement such as: ‘values increased/decreased at an increasing/decreasing rate. Please reword the results section following these indications.
The discussion section needs to be rewritten according to the new results description.
Minor comments
Line 41. Round P values to 3 decimals
Line 47. Provide the actual P value.
Line 83-86. Repeated lines. Either reword or delete
Line 155-156. Could authors be more specific? what did authors analyze?
Line 197-203. These lines are not "basal diet and experimental design"
Table 2. Check abbreviations. There seems to be abbreviation misspelling. As it is now it seems that units are only provided for ingredients, not for chemical composition. Either report OM or ash content, but not both.
Line 228. Authors measure apparent digestibility using an internal marker to estimate fecal output. But this technique still requires knowing intake to estimate digestibility. Did authors measure intake? If so, how? and provide this information in Table results since animals were fed ad libitum.
Line 233. Intake level has a strong effect on digestibility. Did authors find an effect of treatment on intake?
Line 241-243. Authors did not report how they analyzed AIA of feed and feces or any other determination. I suggest adding a chemical determinations section to describe chemical analysis of both in vitro and in vivo trials.
Line 261-262. Disclose what post hoc comparison test was used.
Table 3. Authors measured gas and methane production at different time points. They should indicate at what time they are showing gas and methane production. Delete ANOVA and just leave the P value rounded to 3 decimals.
Line 290-291. Authors performed a polynomial analysis instead of a multiple comparison test. Please delete the superscripts and this comparison in Table. Moreover, it is not reported in the statistical analysis section.
Line 268-269. Authors cannot do these kind of comparisons anymore. Please, refer only to quadartic-linear effects.
Line 271-274. Authors cannot do these kind of comparisons anymore. Please, refer only to quadartic-linear effects.
Line 279-281. Check major comment to describe a linear effect
Line 281. Please reword
Line 281-283. Authors cannot do these kind of comparisons anymore. Please, refer only to quadartic-linear effects.
Line 283-284. Please reword
Line 284-286. Authors cannot do these kind of comparisons anymore. Please, refer only to quadartic-linear effects.
Line 295-297. Authors cannot do these kind of comparisons anymore. Please, refer only to quadartic-linear effects.
Line 297-298. Check major comment to describe a linear effect: linear increase or decrease?
Line 300-301. Authors cannot do these kind of comparisons anymore. Please, refer only to quadartic-linear effects.
Line 303-306. Authors cannot do these kind of comparisons anymore. Please, refer only to quadartic-linear effects.
Line 308-311. Authors cannot do these kind of comparisons anymore. Please, refer only to quadartic-linear effects.
Line 311-314. Check major comment to describe a linear effect.
Line 315-318. Authors cannot do these kind of comparisons anymore. Please, refer only to quadartic-linear effects.
Line 318-319. Check major comment to describe a linear effect: linear increase or decrease?
Line 333-334. Did authors determine the cellulose and hemicellulose content? If this was calculated instead, this should have been stated in the material and methods section.
Line 334-336. Do not describe numerical differences. Report significant effects.
Line 343. I suggest rewording the subsections headings. This one is very similar to that in line 307 and therefore confusing.
Line 346. compared to control and natural zeolite
Line 346-347. This is dicussion. Please reword to describe effects. Were there significant differences among treatments?
Line 348. The comparison between both zeolites is missing.
Line 349-350. Compared to control. The comparison between both zeolites is missing.
Table 7. Authors do not mention in the material and methods section that they measure pH or ammonia at 0 hour, just 3 h after feeding. Either delete or reword the material and methods section. Some abbreviations are missing in the table footnote.
Author Response
Dear Prof. Reviewer (1)
We would like to appreciate you so much for your valuable comments on the Manuscript ID: animals-1216706, which we have taken at our considerations.
Major comments
The presentation of results needs to be improved. When describing a quadratic impact in the text, it is critical that its form be defined. For example, only a significant quadratic effect could lead to a statement such as: ‘values were highest/lowest at the intermediate addition level’, whereas significant linear and quadratic effects could lead to a statement such as: ‘values increased/decreased at an increasing/decreasing rate. Please reword the results section following these indications.
Response: The results section was reworded again according to the new statistical analyses
The discussion section needs to be rewritten according to the new results description.
Response: Reworded
Minor comments
Line 41. Round P values to 3 decimals
Response: done
Line 47. Provide the actual P value.
Response: done
Line 83-86. Repeated lines. Either reword or delete
Response: reworded
Line 155-156. Could authors be more specific? what did authors analyze?
Response: done
Line 197-203. These lines are not "basal diet and experimental design"
Response: it is not clear for us, what do you want
Table 2. Check abbreviations. There seems to be abbreviation misspelling. As it is now it seems that units are only provided for ingredients, not for chemical composition. Either report OM or ash content, but not both.
Response: The abbreviations were checked, units of chemical composition was added, ash was deleted
Line 228. Authors measure apparent digestibility using an internal marker to estimate fecal output. But this technique still requires knowing intake to estimate digestibility. Did authors measure intake? If so, how? and provide this information in Table results since animals were fed ad libitum.
Response: Dry matter intake was determined but the marker technique (lignin) for measuring the nutrients digestibility, which did not require intake so we did not include at the experiment
Anyway, we have included at the table 7
Line 233. Intake level has a strong effect on digestibility. Did authors find an effect of treatment on intake?
Response: no statistical difference was detected
Line 241-243. Authors did not report how they analyzed AIA of feed and feces or any other determination. I suggest adding a chemical determinations section to describe chemical analysis of both in vitro and in vivo trials.
Response: we did not measure AIA as an internal marker but we have used lignin as an internal marker
Line 261-262. Disclose what post hoc comparison test was used.
Response: done
Table 3. Authors measured gas and methane production at different time points. They should indicate at what time they are showing gas and methane production. Delete ANOVA and just leave the P value rounded to 3 decimals.
Response: done
Line 290-291. Authors performed a polynomial analysis instead of a multiple comparison test. Please delete the superscripts and this comparison in Table. Moreover, it is not reported in the statistical analysis section.
Response: done
Line 268-269. Authors cannot do these kind of comparisons anymore. Please, refer only to quadartic-linear effects.
Response: reworded
Line 271-274. Authors cannot do these kind of comparisons anymore. Please, refer only to quadartic-linear effects.
Response: reworded
Line 279-281. Check major comment to describe a linear effect
Response: reworded
Line 281. Please reword
Response: done
Line 281-283. Authors cannot do these kind of comparisons anymore. Please, refer only to quadartic-linear effects.
Response: done
Line 283-284. Please reword
Response: done
Line 284-286. Authors cannot do these kind of comparisons anymore. Please, refer only to quadartic-linear effects.
Response: reworded
Line 295-297. Authors cannot do these kind of comparisons anymore. Please, refer only to quadartic-linear effects.
Response: done
Line 297-298. Check major comment to describe a linear effect: linear increase or decrease?
Response: done
Line 300-301. Authors cannot do these kind of comparisons anymore. Please, refer only to quadartic-linear effects.
Response: done
Line 303-306. Authors cannot do these kind of comparisons anymore. Please, refer only to quadartic-linear effects.
Response: done
Line 308-311. Authors cannot do these kind of comparisons anymore. Please, refer only to quadartic-linear effects.
Response: done
Line 311-314. Check major comment to describe a linear effect.
Response: done
Line 315-318. Authors cannot do these kind of comparisons anymore. Please, refer only to quadartic-linear effects.
Response: done
Line 318-319. Check major comment to describe a linear effect: linear increase or decrease?
Response: done
Line 333-334. Did authors determine the cellulose and hemicellulose content? If this was calculated instead, this should have been stated in the material and methods section.
Response: Cellulose and hemicellulose were calculated, done
Line 334-336. Do not describe numerical differences. Report significant effects.
Response: done
Line 343. I suggest rewording the subsections headings. This one is very similar to that in line 307 and therefore confusing.
Response: done
Line 346. compared to control and natural zeolite
Response: done
Line 346-347. This is dicussion. Please reword to describe effects. Were there significant differences among treatments?
Response: reworded
Line 348. The comparison between both zeolites is missing.
Response: done
Line 349-350. Compared to control. The comparison between both zeolites is missing.
Response: done
Table 7. Authors do not mention in the material and methods section that they measure pH or ammonia at 0 hour, just 3 h after feeding. Either delete or reword the material and methods section. Some abbreviations are missing in the table footnote
Response: reworded and the pH and ammonia were measure at 3 h after feeding; the abbreviations are done
We wish that our responses to your comments will be satisfied
Regards and thanks again
